


**Interactions of organosulfates with water vapor under sub- and supersaturated**
**conditions**
Chao Peng,[1,2] Patricia N. Razafindrambinina,[3] Kotiba A. Malek,[4] Lanxiadi Chen,[1,2,7] Weigang
Wang,[5] Ru-Jin Huang,[6] Yuqing Zhang,[1,2] Xiang Ding,[1,2] Maofa Ge,[5] Xinming Wang,[1,2] Akua A.
Asa-Awuku,[3,4] and Mingjin Tang[1,2,7,*]
[1] State Key Laboratory of Organic Geochemistry, Guangdong Key Laboratory of Environmental
Protection and Resources Utilization, and Guangdong-Hong Kong-Macao Joint Laboratory for
Environmental Pollution and Control, Guangzhou Institute of Geochemistry, Chinese Academy of
Sciences, Guangzhou 510640, China
[2] CAS Center for Excellence in Deep Earth Science, Guangzhou 510640, China
[3] Department of Chemistry and Biochemistry, College of Computer, Mathematical and Natural
Sciences, University of Maryland, College Park, MD 20742, USA
[4] Department of Chemical and Biomolecular Engineering, A. James Clark School of Engineering,
University of Maryland, College Park, MD 20742, USA
[5] State Key Laboratory for Structural Chemistry of Unstable and Stable Species, Beijing National
Laboratory for Molecular Sciences, CAS Research/Education Center for Excellence in Molecular
Sciences, Institute of Chemistry, Chinese Academy of Sciences, Beijing 100190, China
[6] Key Laboratory of Aerosol Chemistry and Physics, State Key Laboratory of Loess and
Quaternary Geology, Institute of Earth and Environment, Chinese Academy of Sciences, Xi'an
710061, China





[7] University of Chinese Academy of Sciences, Beijing 100049, China
*Correspondence: Mingjin Tang (mingjintang@gig.ac.cn)





**Abstract**

Organosulfates (OS) are important constituents of secondary organic aerosols, but their hygroscopic properties and cloud condensation nucleation (CCN) activities have not been well understood. In this work we employed three complementary techniques to characterize interactions of several OS with water vapor under sub- and supersaturated conditions. A vapor sorption analyzer was used to measure mass changes of OS samples with RH (0-90%); among the 11 organosulfates examined, only sodium methyl sulfate (methyl-OS), sodium ethyl sulfate (ethyl-OS), sodium octyl sulfate (octyl-OS) and potassium hydroxyacetone sulfate were found to deliquesce as RH increased, and their mass growth factors at 90% RH were determined to be 3.652±0.064, 3.575±0.024, 1.591±0.004 and 2.202±0.031. Hygroscopic growth of methyl-, ethyl- and octyl-OS aerosols was also studied using a humidity tandem differential mobility analyzer (H-TDMA); continuous hygroscopic growth was observed, and their growth factors at 90% RH were determined to be 1.83±0.03, 1.79±0.02 and 1.21±0.02. We further investigated CCN activities of methyl-, ethyl- and octyl-OS aerosols, and their single hygroscopicity parameters ($\kappa_{\mathrm{ccn}}$) were determined to be 0.459±0.021, 0.397±0.010 and 0.206±0.008. For methyl- and ethyl-OS aerosols, $\kappa_{\mathrm{ccn}}$ values agree reasonably well with those derived from H-TDMA measurements ($\kappa_{\mathrm{gf}}$), whereas $\kappa_{\mathrm{ccn}}$ was found to be significantly larger than $\kappa_{\mathrm{gf}}$ for octyl-OS, likely due to both solubility limit and surface tension reduction.





## 1 Introduction

Secondary organic aerosol (SOA) contributes approximately 70% to the global atmospheric organic aerosols (Hallquist et al., 2009; Jimenez et al., 2009). SOA can affect the Earth's radiative forcing and climate directly by scattering and absorbing solar and terrestrial radiation, and also indirectly by acting as cloud condensation nuclei (CCN) or ice nucleating particles (Moise et al., 2015; Shrivastava et al., 2017). Consequently, it is important to understand the source, formation and physicochemical properties of SOA (Pöschl, 2005; Jimenez et al., 2009; Noziere et al., 2015). However, SOA concentrations on the global scale are significantly underestimated by many modeling studies (Heald et al., 2005; Kanakidou et al., 2005; Ervens et al., 2011; McNeill et al., 2012; Shrivastava et al., 2017), indicating that there might exist unknown while important precursors and/or formation mechanisms of SOA.

Organosulfates (OS), which could contribute to the total mass of ambient organic aerosols by as much as 30%, may largely explain the discrepancy between observed and modeled global SOA budgets (Surratt et al., 2008; Tolocka and Turpin, 2012; Liao et al., 2015). A number of field measurements have observed significant amounts of OS in ambient aerosols in different regions over the globe (Froyd et al., 2010; Kristensen and Glasius, 2011; He et al., 2014; Hettiyadura et al., 2015; Riva et al., 2019; Wang et al., 2019a; Wang et al., 2019b; Zhang et al., 2019; Bruggemann et al., 2020; Wang et al., 2020). For example, the mass concentration of sodium methyl sulfate, the smallest organosulfate, was found to be 0.2-9.3 ng m$^{-3}$ in Centreville, Alabama (Hettiyadura et al., 2017). Hydroxyacetone sulfate, which may originate from both biogenic (Surratt et al., 2008) and anthropogenic emissions (Hansen et al., 2014), has been detected at various locations, such as the Arctic (1.27-9.56 ng m$^{-3}$) (Hansen et al., 2014), Beijing (0.5-7.5 ng m$^{-3}$) (Wang et al., 2018), Xi'an (0.9-2.6 ng m$^{-3}$) (Huang et al., 2018), Centreville (1.5-14.3 ng m$^{-3}$)



(Hettiyadura et al., 2017) and Iowa City ($4.8\pm1.1$ ng m$^{-3}$) (Hughes and Stone, 2019). In addition,
benzyl and phenyl sulfates were also ubiquitous in the troposphere, with reported concentrations
up to almost 1 ng m$^{-3}$ (Kundu et al., 2013; Ma et al., 2014; Staudt et al., 2014; Huang et al., 2018).

As OS are ubiquitous and abundant in the troposphere, it is important to understand their

hygroscopic properties and CCN activities in order to assess their environmental and climatic
effects (Kanakidou et al., 2005; Moise et al., 2015; Tang et al., 2016; Tang et al., 2019a). However,
to our knowledge, only two previous studies have explored their hygroscopic properties and CCN
activities (Hansen et al., 2015; Estillore et al., 2016). Hansen et al. (2015) investigated hygroscopic
growth and CCN activation of limonene-derived OS (with molecular weight of 250 Da) and their
mixtures with ammonium sulfate. Hygroscopicity of pure limonene-derived OS was weak, and its
hygroscopic growth factors were determined to be 1.0 at 80% RH and 1.2 at 93% RH (Hansen et
al., 2015). Estillore et al. (2016) investigated hygroscopic growth of a series of OS, including
potassium salts of glycolic acid sulfate, hydroxyacetone sulfate, 4-hydroxy-2,3-epoxybutane
sulfate, and 2-butenediol sulfate, as well as sodium salts of benzyl sulfate, methyl sulfate, ethyl
sulfate, and propyl sulfate. Continuous hygroscopic growth (i.e. without obvious deliquescence)
was observed for these OS aerosols (Estillore et al., 2016); in addition, their hygroscopic growth
factors at 85% RH were determined to vary between 1.29 and 1.50, suggesting that their
hygroscopicity showed substantial variation. In summary, it is fair to state that hygroscopic
properties and CCN activities of OS have not been well understood.

In this work, three complementary techniques were used to investigate hygroscopic properties

and CCN activities of a series of OS, including sodium methyl sulfate, sodium ethyl sulfate,
sodium octyl sulfate, sodium dodecyl sulfate, potassium hydroxyacetone sulfate, potassium 3-
hydroxy phenyl sulfate, potassium benzyl sulfate, potassium 2-methyl benzyl sulfate, potassium





3-methyl benzyl sulfate, potassium 2,4-dimethyl benzyl sulfate and potassium 3,5-dimethyl benzyl
sulfate. A vapor sorption analyzer was employed to measure mass change of these OS samples as
a function of RH. In addition, hygroscopic growth (change in mobility diameters) and CCN
activation of submicron aerosol particles were studied for sodium methyl sulfate, sodium ethyl
sulfate and sodium octyl sulfate, using a humidity tandem differential mobility analyzer (H-TDMA)
and a cloud condensation nuclei counter (CCNc). Due to their very limited quantities, we could
not carry out H-TDMA and CCNc measurements for other OS samples which were synthesized
by us. In addition, we also investigated the impacts of sodium methyl sulfate, sodium ethyl sulfate
and sodium octyl sulfate on hygroscopic properties and CCN activities of ammonium sulfate.
**2 Experimental section**
**2.1 Chemicals and reagents**

Sodium methyl sulfate ($CH_3SO_4Na$, >98%) and sodium ethyl sulfate ($C_2H_5SO_4Na$, >98%)

were purchased from Tokyo Chemical Industry (TCI); sodium octyl sulfate ($C_8H_{17}SO_4Na$, >99%),
sodium dodecyl sulfate ($C_{12}H_{25}SO_4Na$, >99%) and ammonium sulfate (>99.5%) were supplied by
Aldrich. The other seven OS, including potassium hydroxyacetone sulfate, potassium 3-hydroxy
phenyl sulfate, potassium benzyl sulfate, potassium 2-methyl benzyl sulfate, potassium 3-methyl
benzyl sulfate, potassium 2,4-dimethyl benzyl sulfate and potassium 3,5-dimethyl benzyl sulfate,
were synthesized using the method described by Huang et al. (2018), and their purities were found
to be >95% using nuclear magnetic resonance analysis. Chemical formulas and molecular
structures of OS investigated in this study can be found in Figure 1.



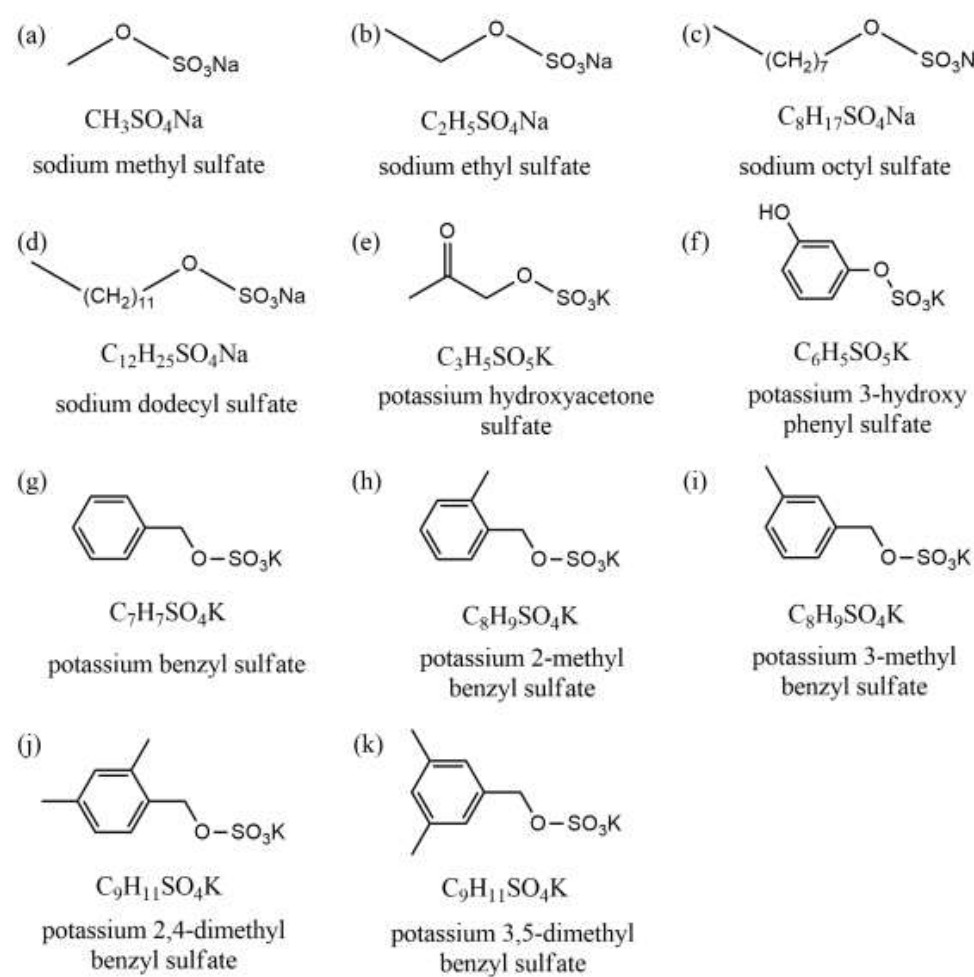

**Figure 1.** Chemical formulas and molecular structures of organosulfates investigated in this study.

## 2.2 VSA experiments

A vapor sorption analyzer (VSA), commercialized by TA Instruments (New Castle, DE, USA), was used to measure mass change of organosulfates as a function of RH. Experimental details can be found in our previous studies (Chen et al., 2019; Gu et al., 2017; Guo et al., 2019; Tang et al., 2019b), and are thus described here briefly. Experiments were conducted at $25 \pm 0.1$ °C





and in the RH range of 0-90%. A high precision balance was used to measure the sample mass at
different RH with a stated sensitivity of <0.1 µg, and the dry mass of samples used in this work
was typically around 1.0 mg.

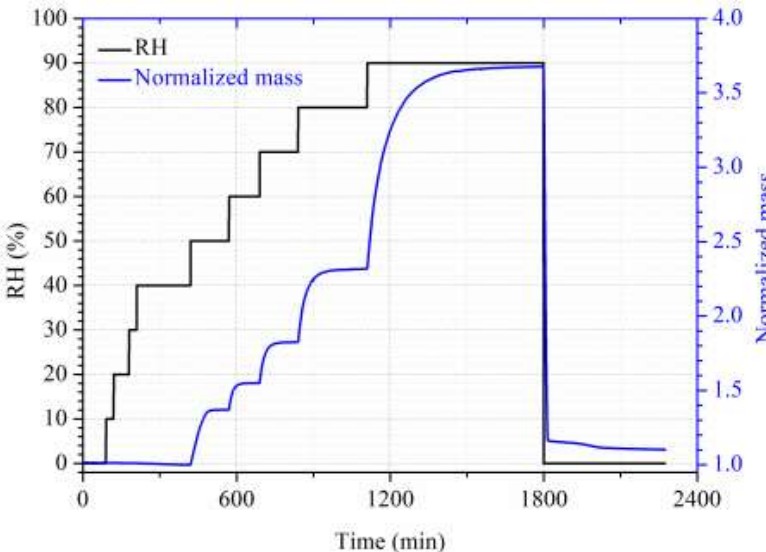


**Figure 2.** Change in RH (black curve, left *y* axis) and normalized sample mass (blue curve, right
*y* axis) of $CH_3SO_4Na$ with of time in a typical vapor sorption analyzer experiment at 25 °C.

As shown in Figure 2, the mass of OS samples at different RH was determined by the VSA

using the following method. RH was set to <1% to dry the sample; after the sample mass was
stabilized, RH was increased to 90% stepwise with an interval of 10% per step; at the end, RH was
changed back to <1% to dry the sample again. The sample was considered to reach the equilibrium
at a given RH when the mass change was measured to be <0.1% within 30 min. All the experiments
were conducted at least three times in our work. The sample mass at a given RH (*m*) was
normalized to that at <1% RH ($m_0$) to determine the mass growth factor, defined as $m/m_0$.


### 2.3 H-TDMA experiments

A custom-built hygroscopicity tandem differential mobility analyzer (H-TDMA) was used to measure the mobility diameters of OS aerosol particles at different RH (5-90%) at $24 \pm 1$ °C. The instrument was detailed elsewhere (Jing et al., 2016; Peng et al., 2016), and therefore only a brief introduction is given here. A commercial atomizer (MSP 1500) was used to produced polydisperse aerosol particles from dilute OS solutions in water (around 0.1 wt %), and the generated aerosol was dried to <5% RH by passing the aerosol flow through a Nafion dryer (MD-110-12S) and then a silica gel diffusion dryer. The dry aerosol flow was subsequently split to two flows. One aerosol flow was sent to the vent, and the other aerosol flow (0.3 L min$^{-1}$) was passed through the first differential mobility analyzer (DMA) to produce quasi-monodisperse aerosol particles with a mobility diameter of 100 nm. After that, the aerosol flow was humidified to a desired RH by flowing through a humidification section, which was made of two Nafion tubes (MD-700-12F-1) connected in series, and the residence time in the humidification section was ~27 s. Finally, the size distribution of humidified aerosol was measured by a scanning mobility particle sizer (SMPS), which consisted of the second DMA coupled to a condensation particle counter (CPC 3776, TSI). RH of the aerosol flow and the sheath flow in the second DMA were maintained to be equal and monitored using a commercial dew-point hygrometer (Michell, UK) with a stated uncertainty of ±0.08% RH. In addition, the flow rate ratio of the sheath flow to the aerosol flow was set to 10:1 for both DMAs.

Hygroscopic growth factors (GF), defined as $d/d_0$ ($d$ is the mobility diameter at a given RH and $d_0$ is the mobility diameter at RH <5%) were reported. All the experiments were conducted in triplicate. During our experiments, ammonium sulfate was used to calibrate the H-TDMA system



routinely, and the absolute differences between the measured and theoretical GF at 90% RH were
found to be within 0.04, confirming the robustness of our measurements.
**2.4 CCN experiments**

The CCN activity of aerosol particles was determined using a commercial cloud condensation

nuclei counter (CCNc, CCN-100, Droplet Measurement Technologies, Longmont, CO, USA)
described in previous studies (Roberts and Nenes, 2005; Lance et al., 2006; Moore et al., 2010).
Polydisperse aerosol particles were generated using a commercial atomizer (TSI 3076), in which
concentrations of solutions used were around 0.1 g/L. The wet aerosol flow generated was passed
through two silica gel diffusion dryers to reduce its RH to <5% RH. After that, a dry aerosol flow
(~800 mL min$^{-1}$) was passed through a DMA (TSI 3081) in size scanning mode to produce quasi-
monodisperse aerosols, and subsequently the aerosol flow was split to two streams: one stream
(~300 mL/min) was sampled into a commercial CPC (TSI 3775) to measure total number
concentrations of aerosol particles ([CN]), and the second flow (~500 mL/min) was sampled into
the cloud condensation nuclei counter (CCNc, CCN-100) to measure number concentrations of
CCN ([CCN]).

Activation fractions ([CCN]/[CN]) of size-resolved dry particles were determined using the

Scanning Mobility CCN Analysis (SMCA) method described elsewhere (Moore et al., 2010). In
brief, the DMA was operated in the scanning voltage mode, and thus one activation curve
(activation fractions as a function of dry diameter) could be obtained in 60-120s. The multiple
charge effect was also corrected in this method, and in our work the supersaturation (*SS*) was set
in the range of 0.45-1.13% with the stated uncertainty to be ±0.01%. As shown in Figure 3,
activation fractions of sodium methyl sulfate (methyl-OS) and its internally mixed aerosol with
ammonium sulfate were measured at four different SS with dry mobility diameters between 20



and 100 nm. Activation fractions were fitted versus dry diameters, and the critical particle diameter
($d_{50}$) was determined as the dry diameter at which the activation fraction is equal to 0.5. During
our measurements, ammonium sulfate was used to calibrate supersaturations, and the Pitzer-ion
interaction model was applied in the calibration procedure to account for incomplete dissociation
of ammonium sulfate at droplet activation (Pitzer and Mayorga, 1973; Clegg and Brimblecombe,
1988). The corrected supersaturations were reported in our work.

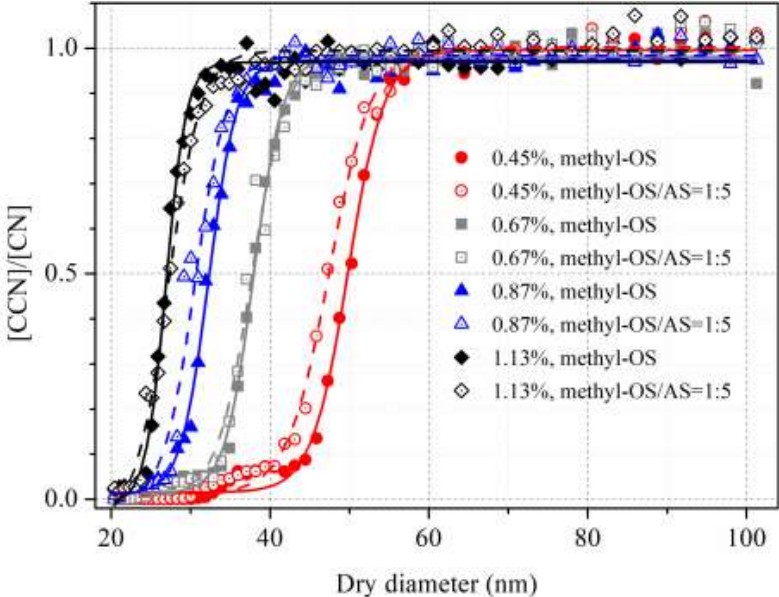


**Figure 3.** Activation fractions of methyl-OS and its internally mixed aerosol particles with
ammonium sulfate (AS) as a function of dry particle diameter at four supersaturations.

## 3 Results and discussion

### 3.1 Mass growth of organosulfates

Figure 4 displays mass growth factors of sodium methyl sulfate (methyl-OS), sodium ethyl

sulfate (ethyl-OS), sodium octyl sulfate (octyl-OS), sodium dodecyl sulfate (dodecyl-OS) and





potassium hydroxyacetone sulfate, and the data are also listed in Table 1. Figure 4a suggests that
methyl-OS was deliquesced when RH was increased from 40% to 50%, and after that mass growth
factors increased further with RH. The mass of ethyl-OS was moderately increased (by ~11%)
when RH was increased from 40% to 50%, and further increase in RH to 60% led to additional
while small increase in sample mass (by ~2%); the increase in sample mass at 50% and 60% RH
may be because ethyl-OS were partially deliquesced at this stage. When RH was increased to 70%,
ethyl-OS was completely deliquesced, and further increase in RH (to 80% and 90%) resulted in
further increase in sample mass. Octyl-OS was only deliquesced when RH was increased from 80%
to 90%, whereas no significant water uptake was observed for dodecyl-OS even at 90% RH. The
mass growth factors at 90% RH were determined to be 3.652±0.064, 3.575±0.024 and 1.591±0.004
for methyl-OS, ethyl-OS and octyl-OS, respectively.

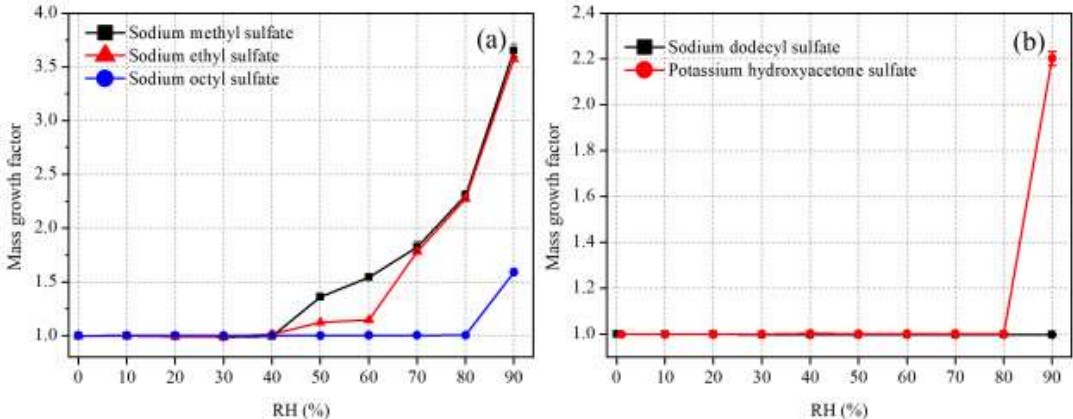


**Figure 4.** Mass growth factors of (a) methyl-, ethyl- and octyl-OS and (b) dodecyl-OS and
potassium hydroxyacetone sulfate as a function of RH at 25 °C. Please note that error bars are
included, but they are too small to be clearly visible.

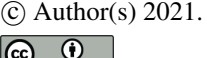



Mass growth factors of seven potassium organosulfates were also investigated, including
potassium hydroxyacetone sulfate, potassium 3-hydroxy phenyl sulfate, potassium benzyl sulfate,
potassium 2-methyl benzyl sulfate, potassium 3-methyl benzyl sulfate, potassium 2,4-dimethyl
benzyl sulfate and potassium 3,5-dimethyl benzyl sulfate. All the compounds did not show obvious
water uptake at 80% RH. When RH was increased to 90%, as shown in Figure 4b, a significant
increase in mass was observed for potassium hydroxyacetone sulfate particles, suggesting that
occurrence of deliquescence, and the mass growth factor was determined to be 2.202±0.031 at 90%
RH. No significant water uptake was observed for the other six potassium organosulfates even
when RH was increased to 90%. We should mention that occasionally small increase in sample
mass (up to 10-20%) was observed for a few samples when RH was increased from 80% to 90%,
and such small increase in sample mass may be caused by water uptake of impurities (such as
potassium hydroxide) contained in these synthesized compounds.

**Table 1.** Mass growth factors ($m/m_0$) and water-to-solute ratios (WSRs) as a function of RH (10-
90 %) at 25 °C for sodium methyl sulfate, sodium ethyl sulfate, sodium octyl sulfate and potassium
hydroxyacetone sulfate. All the errors given in this work are standard deviations.

| RH (%) | sodium methyl sulfate | | sodium ethyl sulfate | |
|---|---|---|---|---|
| | $m/m_0$ | WSR | $m/m_0$ | WSR |
| 10 | 1.000±0.003 | - | 1.000±0.001 | - |
| 20 | 1.000±0.003 | - | 0.992±0.012 | - |
| 30 | 0.989±0.013 | - | 0.991±0.012 | - |
| 40 | 0.996±0.018 | - | 1.015±0.031 | - |
| 50 | 1.360±0.018 | 2.68±0.04 | 1.125±0.005 | - |
| 60 | 1.545±0.034 | 4.06±0.09 | 1.147±0.004 | - |
| 70 | 1.827±0.051 | 6.16±0.17 | 1.785±0.021 | 6.46±0.07 |





| RH (%) | | | 2.306±0.042 | 9.73±0.18 | 2.274±0.024 | 10.48±0.11 |
|---|---|---|---|---|---|---|

| 80 | 2.306±0.042 | 9.73±0.18 | 2.274±0.024 | 10.48±0.11 |
|---|---|---|---|---|
| 90 | 3.652±0.064 | 19.75±0.34 | 3.575±0.024 | 21.19±0.14 |

| RH (%) | sodium octyl sulfate | | potassium hydroxyacetone sulfate | |
|---|---|---|---|---|
| | $m/m_0$ | WSR | $m/m_0$ | WSR |
| 10 | 1.001±0.001 | - | 1.000±0.001 | - |
| 20 | 1.002±0.001 | - | 1.000±0.001 | - |
| 30 | 1.002±0.001 | - | 1.000±0.001 | - |
| 40 | 1.002±0.001 | - | 1.003±0.005 | - |
| 50 | 1.003±0.001 | - | 1.002±0.003 | - |
| 60 | 1.003±0.001 | - | 1.002±0.004 | - |
| 70 | 1.004±0.001 | - | 1.003±0.003 | - |
| 80 | 1.005±0.001 | - | 1.002±0.004 | - |
| 90 | 1.591±0.004 | 7.63±0.02 | 2.202±0.031 | 12.84±0.18 |


For deliquesced samples, measured mass changes can be converted to water to solute ratios
(WSRs), defined as the molar ratio of $H_2O$ to sulfur. The WSRs data are summarized in Table 1
for sodium methyl sulfate, sodium ethyl sulfate, sodium octyl sulfate and potassium
hydroxyacetone sulfate. As shown in Table 1, WSRs at 90% RH were determined to be 19.75±0.34,
21.19±0.14, 7.63±0.02 and 12.84±0.18 for sodium methyl sulfate, sodium ethyl sulfate, sodium
octyl sulfate and potassium hydroxyacetone sulfate at 25 ℃.

**3.2 Hygroscopic growth of aerosols**

**3.2.1 Organosulfates**

H-TDMA was employed to measure hygroscopic growth factors of 100 nm methyl-, ethyl-
and octyl-OS aerosols as a function of RH (up to 90%), and the results are shown in Figure 5 and
Table 2. In addition, no significant hygroscopic growth was observed for dodecyl-OS for RH up
to 90%. We did not investigate hygroscopic growth of other OS aerosols due to the very small
quantity of these synthesized compounds.





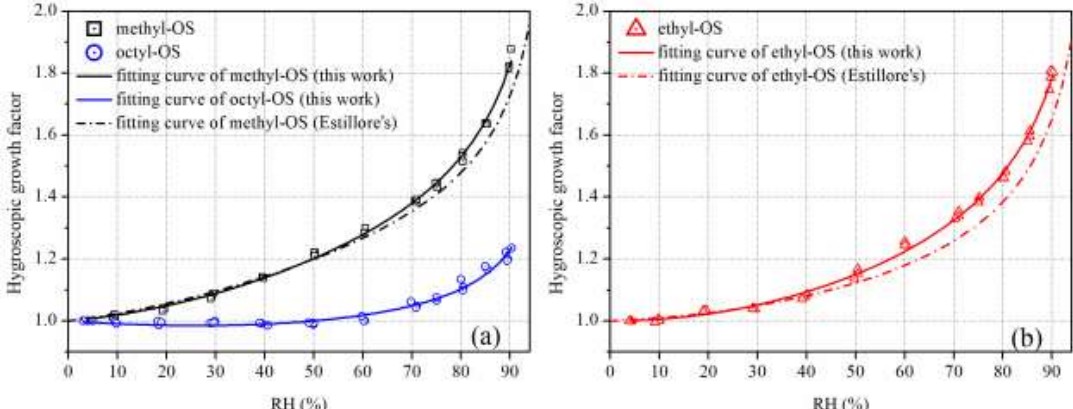


**Figure 5.** Hygroscopic growth factors of (a) methyl- and octyl-OS and (b) ethyl-OS aerosols as a

function of RH. Solid curves represent fitted curves in our work using Eq. (1). For comparison,

the fitted curves reported by Estillore et al. (2016) are presented by dashed curves.

As shown in Figure 5, methyl-, ethyl- and octyl-OS aerosols all exhibited continuous

hygroscopic growth without obvious phase transitions. To our knowledge, only one previous study

(Estillore et al., 2016) investigated hygroscopic growth of methyl- and ethyl-OS aerosols using a

H-TDMA, and continuous hygroscopic growth was also observed. The continuous growth

behavior can be attributed to the amorphous state of aerosol particles, which would take up water

at very low RH. For methyl-OS aerosol, GFs were determined in our work to be 1.53±0.01,

1.63±0.01 and 1.83±0.03 at 80%, 85% and 90% RH; for comparison, its GF was measured to be

1.50 at 85% RH by Estillore et al. (2016), only ~8% smaller than our result. In our work, GFs were

determined to be 1.47±0.01, 1.60±0.02 and 1.79±0.02 at 80%, 85% and 90% RH for ethyl-OS

aerosol; for comparison, it was measured to be 1.45 at 85% RH in the previous study (Estillore et

al., 2016), only ~9% smaller than our result. Overall, our measured GFs agree very well with those

reported by Estillore et al. (2016) for methyl- and ethyl-OS, while the highest RH we reached was





90%, compared to 85% by Estillore et al. (2016). With respect to octyl-OS aerosol, GF were
determined to be 1.11±0.02, 1.17±0.01 and 1.21±0.02 at 80%, 85% and 90% RH in our work; to
our knowledge, hygroscopic growth of octyl-OS aerosol has not been explored previously.
Compared to ammonium sulfate (1.753 at 90% RH), GFs at 90% RHs were found to be slightly
larger for methyl- and ethyl-OS, but significantly smaller for octyl-OS.

**Table 2.** Hygroscopic growth factors (GFs) of methyl-, ethyl- and octyl-OS aerosols at different
RH. All the errors given in this work are standard deviations.

| RH (%) | sodium methyl sulfate | sodium ethyl sulfate | sodium octyl sulfate |
|---|---|---|---|
| 5 | 1.00±0.01 | 1.00±0.01 | 1.00±0.01 |
| 10 | 1.01±0.01 | 1.00±0.01 | 1.00±0.01 |
| 20 | 1.04±0.01 | 1.03±0.01 | 0.99±0.01 |
| 30 | 1.08±0.01 | 1.04±0.01 | 1.00±0.01 |
| 40 | 1.14±0.01 | 1.08±0.01 | 0.99±0.01 |
| 50 | 1.22±0.01 | 1.15±0.01 | 0.99±0.01 |
| 60 | 1.29±0.01 | 1.25±0.01 | 1.01±0.01 |
| 70 | 1.39±0.01 | 1.34±0.01 | 1.05±0.01 |
| 75 | 1.44±0.01 | 1.39±0.01 | 1.07±0.01 |
| 80 | 1.53±0.01 | 1.47±0.01 | 1.11±0.02 |
| 85 | 1.63±0.01 | 1.60±0.02 | 1.17±0.01 |
| 90 | 1.83±0.03 | 1.79±0.02 | 1.21±0.02 |


When aerosol particles take up water continuously, the RH-dependent GFs can usually be
fitted using Eq. (1) (Kreidenweis et al., 2005):
$$GF = [1 + (a + b \cdot \frac{RH}{100} + c \cdot (\frac{RH}{100})^2) \cdot \frac{RH}{100-RH}]^{1/3} \quad (1)$$



where $a$, $b$ and $c$ are coefficients obtained from fitting using Eq. (1). As shown in Figure 5,
hygroscopic growth factors of methyl-, ethyl- and octyl-OS aerosols can be fitted by Eq. (1), and
the obtained coefficients ($a$, $b$ and $c$) are summarized in Table 3.

**Table 3.** The three coefficients ($a$, $b$ and $c$) obtained by using Eq. (1) to fit RH-dependent GFs for
sodium methyl sulfate, sodium ethyl sulfate and sodium octyl sulfate aerosols.

| organosulfates | $a$ | $b$ | $c$ |
|---|---|---|---|
| sodium methyl sulfate | 0.42182 | 1.20336 | -1.15508 |
| sodium ethyl sulfate | 0.00174 | 1.61805 | -1.15502 |
| sodium octyl sulfate | -0.31868 | 0.86233 | -0.44623 |


**3.2.2 Comparison between VSA and H-TDMA measurements**
Figure 4 shows that obvious deliquescence transitions were observed for methyl-, ethyl-, and
octyl-OS in the VSA experiments; in contrast, as revealed by Figure 5, continuous hygroscopic
growth without obvious phase transitions was observed for methyl-, ethyl- and octyl-OS aerosol
particles in H-TDMA measurements, suggesting that these aerosol particles may exist in
amorphous state. Estillore et al. (2016) employed a H-TDMA to investigate hygroscopic properties
of several OS aerosols, and similarly they found that those aerosols, including methyl-OS, ethyl-
OS and potassium hydroxyacetone sulfate which were also examined in our work, displayed
continuous hygroscopic growth.
For completely deliquesced particles, if it is assumed that the particle is spherical and that the
particle volume at a given RH is equal to the sum of the dry particle volume and the volume of





particulate water, particle mass change, measured using the VSA, can then be converted to
hygroscopic GF, using Eq. (2):
$$GF = \sqrt[3]{1 + (\frac{m}{m_0} - 1) \cdot \frac{\rho_0}{\rho_w}} \quad (2)$$

where $\rho_0$ and $\rho_w$ are the density of the dry sample and water, respectively. The density of methyl-,
ethyl- and octyl-OS particles were reported to be 1.60, 1.46 and 1.19 g cm$^{-3}$ with an uncertainty of
20-30% (Kwong et al., 2018; ChemistryDashboard, 2021). Figure 6 compares VSA-derived GFs
and those measured using H-TDMA for methyl-, ethyl- and octyl-OS, and it can be concluded that
for RH at which samples used in the VSA experiments were deliquesced, GFs derived from mass
change measured using VSA agree relatively well with those directly measured using H-TDMA.
For example, at 90% RH GFs were measured by H-TDMA to be 1.83±0.03, 1.79±0.02 and
1.21±0.02 for methyl-, ethyl- and octyl-OS, while at the same RH their GFs derived from VSA
measurements were found to be 1.74±0.01, 1.68±0.01 and 1.19±0.01, only 6% (or less) smaller
than those measured using H-TDMA.

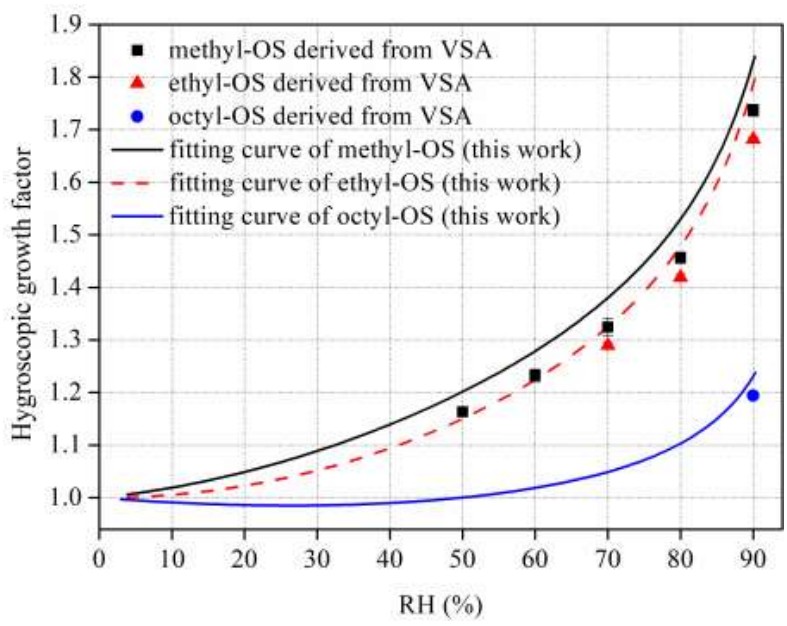




**Figure 6.** Comparison between hygroscopic GFs of methyl-, ethyl- and octyl-OS derived from VSA experiments to those measured using H-TDMA. Please note that H-TDMA results are presented as the three-parameter curves obtained. Error bars are included, but they are too small to be clearly visible.

### 3.2.3 Internally mixed aerosols

We also investigated hygroscopic properties of methyl-, ethyl- and octyl-OS aerosols internally mixed with ammonium sulfate (AS), and the results are summarized in Table 4. Figure 7a displays GFs of 100 nm methyl-OS/AS mixed aerosols with mass ratios of 1:1 and 1:5. The 1:1 mixed aerosol particle showed a deliquescence transition at 70% RH, while the 1:5 mixed aerosols showed a deliquescence transition at 75% RH, which was lower than the deliquescence RH (DRH, 80%) of AS. Here the DRH is defined as the RH at which the mixed aerosols are completely deliquesced (Choi and Chan, 2002). Figure 7a suggests that before full deliquescence, significant hygroscopic growth was also observed, i.e. pre-deliquescence of mixed particles occurred when RH was lower than their DRH. Pre-deliquescence was widely reported in previous studies which investigated hygroscopic properties of inorganic/organic mixed aerosols (Choi and Chan, 2002; Prenni, 2003; Wise et al., 2003; Brooks, 2004; Marcolli and Krieger, 2006; Wu et al., 2011; Lei et al., 2014; Jing et al., 2016; Estillore et al., 2017). For example, Choi and Chan. (2002) investigated hygroscopic behaviors of internal mixed particles which consisted of water-soluble organic compounds and AS, and found that the internal mixing with organics (such as malonic and citric acids) could reduce the DRH of AS, due to the ability of organics to absorb water at low RH.

Internal mixing with ethyl- and octyl-OS could also reduce the DRH of AS. As shown in Figure 7b, ethyl-OS/AS mixed aerosols were deliquesced at 70% RH when the mass ratio of ethyl-

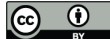



OS to AS was 1:1 and at 80% RH when the mass ratio was 1:5. In addition, Figure 7c suggested
that the deliquescence of octyl-OS/AS aerosols took place at 75% RH for the 1:1 mixture and at
80% for 1:5 mixture.

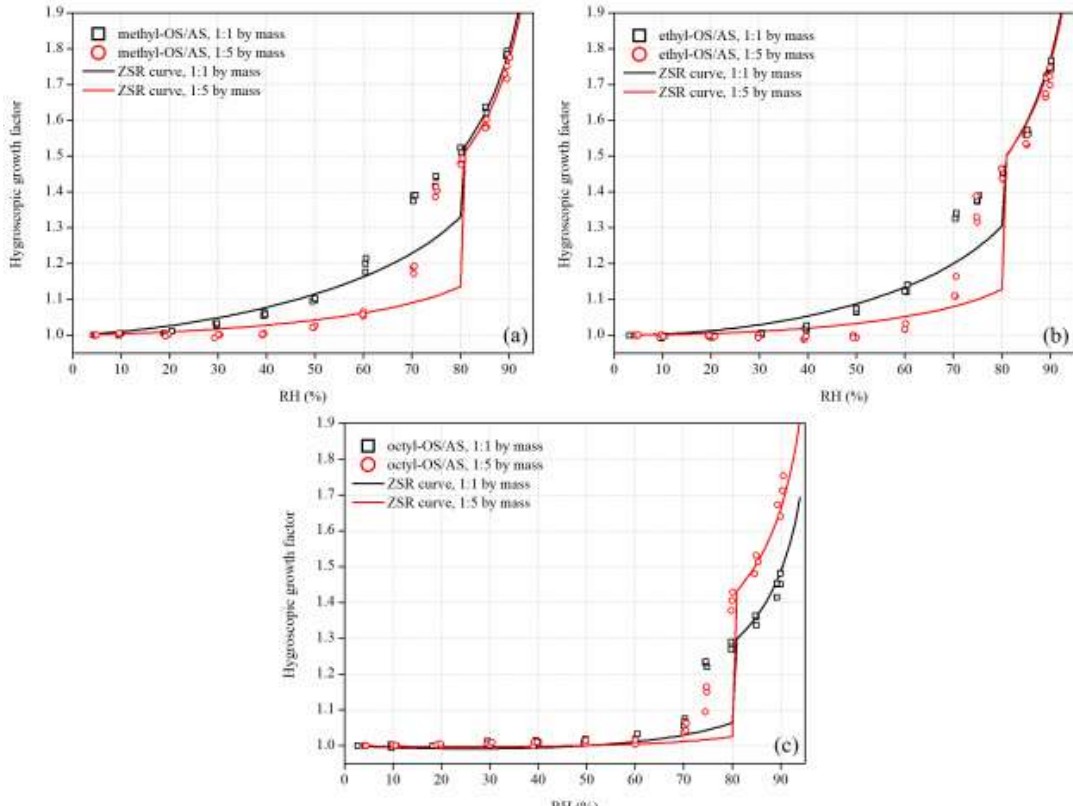


**Figure 7.** Hygroscopic growth factors of (a) methyl-OS/AS, (b) ethyl-OS/AS, and (c) octyl-OS/AS
aerosols as a function of RH. The mass ratios of methyl-, ethyl-, and octyl-OS to AS were 1:1 and
1:5, respectively. Solid curves represent hygroscopic growth factors of mixed aerosols predicted
using the ZSR method.

**Table 4.** Hygroscopic GF of methyl-, ethyl, and octyl-OS internally mixed with AS (their mass
ratios are 1:1 and 1:5) at different RH. All the errors given in this work are standard deviations.





| RH (%) | methyl-OS/AS | | ethyl-OS/AS | | octyl-OS/AS | |
|---|---|---|---|---|---|---|
| | 1:1 | 1:5 | 1:1 | 1:5 | 1:1 | 1:5 |
| 5 | 1.00±0.01 | 1.00±0.01 | 1.00±0.01 | 1.00±0.01 | 1.00±0.01 | 1.00±0.01 |
| 10 | 1.00±0.01 | 1.00±0.01 | 1.00±0.01 | 1.00±0.01 | 1.00±0.01 | 1.00±0.01 |
| 20 | 1.01±0.01 | 1.00±0.01 | 1.00±0.01 | 1.00±0.01 | 1.00±0.01 | 1.00±0.01 |
| 30 | 1.03±0.01 | 1.00±0.01 | 1.00±0.01 | 1.00±0.01 | 1.01±0.01 | 1.01±0.01 |
| 40 | 1.06±0.01 | 1.00±0.01 | 1.02±0.01 | 0.99±0.01 | 1.01±0.01 | 1.01±0.01 |
| 50 | 1.10±0.01 | 1.02±0.01 | 1.07±0.01 | 0.99±0.01 | 1.01±0.01 | 1.01±0.01 |
| 60 | 1.20±0.02 | 1.06±0.01 | 1.13±0.01 | 1.02±0.01 | 1.02±0.01 | 1.01±0.01 |
| 70 | 1.38±0.01 | 1.19±0.01 | 1.33±0.01 | 1.13±0.03 | 1.07±0.01 | 1.05±0.01 |
| 75 | 1.43±0.01 | 1.40±0.01 | 1.38±0.01 | 1.34±0.04 | 1.23±0.01 | 1.14±0.03 |
| 80 | 1.52±0.01 | 1.48±0.01 | 1.46±0.01 | 1.45±0.02 | 1.28±0.01 | 1.40±0.02 |
| 85 | 1.63±0.01 | 1.59±0.01 | 1.56±0.01 | 1.54±0.02 | 1.35±0.02 | 1.51±0.02 |
| 90 | 1.79±0.01 | 1.74±0.02 | 1.74±0.02 | 1.72±0.02 | 1.47±0.02 | 1.69±0.04 |


The Zdanovskii-Stokes-Robinson (ZSR) method (Stokes and Robinson, 1966) has been
widely used to predict hygroscopic growth of internally mixed aerosol particles, assuming that the
interaction among individual species are negligible and that individual species in the mixed
particles take up water independently. According to the ZSR method, GF of a mixed particle, $GF_{mix}$,
can be calculated using Eq. (3) (Malm and Kreidenweis, 1997):
$$GF_{mix} = \sqrt[3]{\sum(\varepsilon_i \cdot GF_i^3)} \quad (3)$$

where $GF_i$ is the GF of $i$th species the dry mixed particle contains. The volume fraction of the $i$th
species in the dry mixed particle, $\varepsilon_i$, can be calculated using Eq. (4):
$$\varepsilon_i = \frac{m_i/\rho_i}{\sum(m_i/\rho_i)} \quad (4)$$

where $m_i$ and $\rho_i$ are the mass fraction and density of the $i$th species. GFs of pure OS, measured in
our work using H-TDMA and presented in Section 3.2.1, and GFs of AS, calculated using the E-





AIM model (Clegg et al., 1998; Wexler and Clegg, 2002), were used as input to predict GFs of
methyl-, ethyl- and octyl-OS internally mixed with AS. Comparisons between measured and
predicted GFs are displayed in Figure 7 for OS/AS mixed aerosols.

As shown in Figure 7a, GFs of methyl-OS/AS mixed aerosols (both the 1:1 and 1:5 mixtures)

could be well predicted using the ZSR method when RH was <60% or >80%, while the ZSR
method underestimated their GFs at 70% and 75% RH. Such underestimation at 70% and 75% RH
is likely to due to that inorganic compounds (AS, in our work) may dissolve partially in the
organics/water solution (which can be formed at much lower RH due to continuous water uptake
of organics) before the mixed particle is completely deliquesced (Svenningsson et al., 2006;
Zardini et al., 2008; Wu et al., 2011); in contrast, the ZSR method assumes that individual species
take up water independently. As shown in Figure 7, the ZSR method also underestimated GFs at
70 and 75% RH for ethyl-OS/AS and octyl-OS/AS mixed aerosols, though good agreement
between measurement and prediction was found at other RH.
**3.3 Cloud condensation nucleation activities**

Figures 3, S2 and S3 show CCN activation curves obtained at four supersaturations for

methyl-, ethyl- and octyl-OS aerosols and their internal mixtures with ammonium sulfate. Each
activation curve was fitted using a Boltzmann sigmoid function to derive the corresponding critical
particle diameter ($d_{50}$), which was then used to calculate $\kappa_{ccn}$ using Eqs. (5a-5b) (Petters and
Kreidenweis, 2007):
$$\kappa_{ccn} = \frac{4A^3}{27 d_{50}^3 \ln^2 S_c} \quad (5a)$$
$$A = \frac{4\sigma_{s/a} M_w}{RT \rho_w} \quad (5b)$$
where $S_c$ is the critical saturation ratio (1+$SS$) of water; $d_{50}$ is the critical particle diameter; $A$ is a
constant which describes the Kelvin effect on a curved surface of a droplet, and depends on the



surface tension ($\sigma_{s/a}$), molecular weight ($M_w$), density ($\rho_w$) of water, temperature ($T$) and the
universal gas constant ($R$). Table 5 summarizes critical diameters at different supersaturations for
aerosol particles examined in this work and their $\kappa_{ccn}$ values.

**Table 5.** Single hygroscopicity parameters derived from hygroscopic growth ($\kappa_{gf}$) and CCN
activity measurements ($\kappa_{ccn}$) for methyl-, ethyl- and octyl-OS and their internal mixtures with
ammonium sulfate (AS). All errors given were standard deviations.

| aerosol | mass ratio | $SS$ (%) | $d_{50}$ (nm) | $\kappa_{ccn}$ | average $\kappa_{ccn}$ | $\kappa_{gf}$ |
|---|---|---|---|---|---|---|
| methyl-OS | - | 0.45 | 52.9±0.9 | 0.432-0.477 | 0.459±0.021 | 0.537-0.604 |
| | - | 0.67 | 41.1±0.8 | 0.416-0.468 | | |
| | - | 0.87 | 33.3±0.4 | 0.471-0.507 | | |
| | - | 1.13 | 28.8±0.5 | 0.431-0.477 | | |
| methyl-OS/AS | 1:5 | 0.45 | 51.9±0.5 | 0.467-0.492 | 0.453±0.027 | 0.454-0.495 |
| | 1:5 | 0.67 | 41.6±0.4 | 0.411-0.436 | | |
| | 1:5 | 0.87 | 33.7±0.5 | 0.453-0.490 | | |
| | 1:5 | 1.13 | 29.2±0.6 | 0.412-0.464 | | |
| ethyl-OS | - | 0.45 | 55.5±0.8 | 0.375-0.410 | 0.397±0.010 | 0.505-0.548 |
| | - | 0.67 | 42.8±0.6 | 0.376-0.406 | | |
| | - | 0.87 | 35.3±0.5 | 0.395-0.428 | | |
| | - | 1.13 | 30.2±0.3 | 0.382-0.408 | | |
| ethyl-OS/AS | 1:5 | 0.45 | 52.3±1.2 | 0.437-0.504 | 0.458±0.024 | 0.435-0.474 |
| | 1:5 | 0.67 | 41.0±0.5 | 0.426-0.459 | | |
| | 1:5 | 0.87 | 33.4±0.6 | 0.463-0.512 | | |
| | 1:5 | 1.13 | 29.2±0.6 | 0.409-0.462 | | |
| octyl-OS | - | 0.45 | 70.0±1.2 | 0.186-0.207 | 0.206±0.008 | 0.076-0.096 |
| | - | 0.67 | 53.2±0.6 | 0.196-0.211 | | |
| | - | 0.87 | 44.1±0.7 | 0.202-0.221 | | |
| | - | 1.13 | 37.1±0.8 | 0.200-0.227 | | |
| octyl-OS/AS | 1:5 | 0.45 | 53.7±0.9 | 0.413-0.456 | 0.436±0.009 | 0.388-0.464 |





| | | | |
|---|---|---|---|
| 1:5 | 0.67 | 41.1±0.5 | 0.426-0.458 |
| 1:5 | 0.87 | 34.4±0.5 | 0.427-0.462 |
| 1:5 | 1.13 | 29.5±0.2 | 0.413-0.434 |


As shown in Table 5, $\kappa_{ccn}$ values were determined to be 0.459±0.021, 0.397±0.010 and
0.206±0.008 for methyl-, ethyl- and octyl-OS, decreasing with alkyl chain length, and this suggests
that the addition of hydrophobic hydrocarbon functional groups to OS reduced their hygroscopicity.
In addition, we investigated CCN activities of alkyl-OS/AS mixed aerosols with a mass ratio of
1:5, and $\kappa_{ccn}$ values were determined to be 0.453±0.027, 0.458±0.024 and 0.436±0.009 for methyl-
OS/AS, ethyl-OS/AS and octyl-OS/AS.

**3.3.1 Comparison between H-TDMA and CCN activities measurements**

It is suggested that the single hygroscopicity parameter, $\kappa$, could describe aerosol-water
interactions under both sub- and supersaturated conditions (Petters and Kreidenweis, 2007). The $\kappa$
values derived from CCN activity measurements, $\kappa_{ccn}$, have been illustrated above; the $\kappa$ values
derived from H-TDMA measurements, $\kappa_{gf}$, can be calculated using Eq. (6) (Petters and
Kreidenweis, 2007; Tang et al., 2016):

$$\kappa_{gf} = (GF^3 - 1)\frac{1-RH}{RH} \quad (6)$$

In this work GF measured at 90% RH were used to calculate $\kappa_{gf}$ values, which are also listed in
Table 5.
Figure 8 compares $\kappa_{ccn}$ and $\kappa_{gf}$ values for the six types of aerosol particles examined. No
significant difference was observed between $\kappa_{gf}$ and $\kappa_{ccn}$ for five types of aerosol particles, and the
relative differences between $\kappa_{ccn}$ and $\kappa_{gf}$ values do not exceed 25%. However, octyl-OS appears to
be an exception, and the average $\kappa_{ccn}$ value (0.206) was ~1.4 times larger than the average $\kappa_{gf}$ value

(0.086).





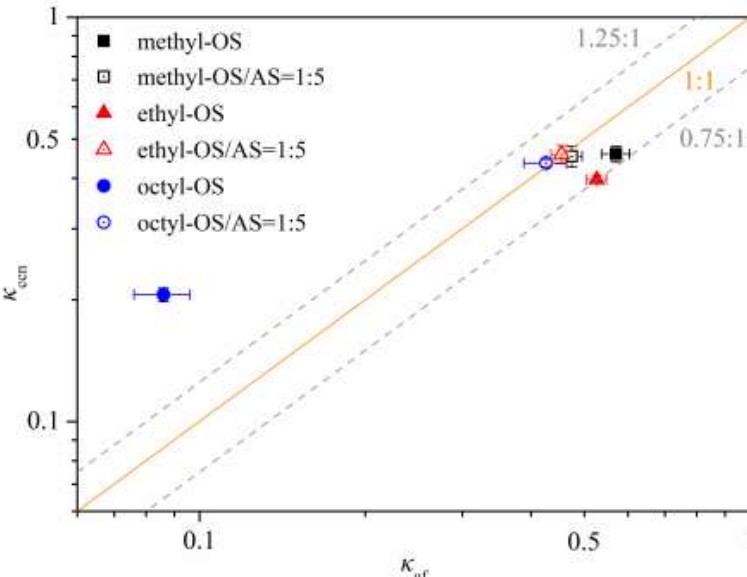


**Figure 8.** Comparison of $\kappa$ values derived from hygroscopic growth ($\kappa_{gf}$) with these derived from

CCN activities ($\kappa_{ccn}$) for methyl-, ethyl- and octyl-OS aerosols as well as their internal mixtures

with ammonium sulfate (the mass ratio was 1:5).


Significant differences between $\kappa_{gf}$ and $\kappa_{ccn}$ were reported in previous studies (Petters et al.,

2009; Wex et al., 2009; Hansen et al., 2015), attributed to several factors discussed below. Petters

and Kreidenweis. (2008) demonstrated that cloud droplet activation was highly sensitive to the

solubility for sparingly soluble compounds in the range of $5 \times 10^{-4}$-$2 \times 10^{-1}$, expressed as volume of

solute per unit volume of water (Petters and Kreidenweis, 2008). Compared to the highly soluble

methyl- and ethyl-OS, the solubility of octyl-OS ($8.43 \times 10^{-4}$-$4.26 \times 10^{-2}$) (Chemistry Dashboard,

2021) is rather limited, and incomplete dissolution at subsaturated condition in H-TDMA

measurements may lead to underestimation of $\kappa_{gf}$ values for octyl-OS; as a result, the solubility

limit may explain the observed difference between $\kappa_{gf}$ and $\kappa_{ccn}$ for octyl-OS. Furthermore, surface





tension is a key factor to influence critical supersaturations at which aerosol particles are activated
to cloud droplets (Petters and Kreidenweis, 2013). We measured surface tensions of alkyl-OS and
alkyl-OS/AS (the mass ratio was 1:5) solutions, and the results are shown in Table S1 and Figure
S4. The surface tension of octyl-OS is much lower than that of pure water, leading to significant
reduction in critical supersaturations and thus overestimation of its $\kappa_{ccn}$ value; for comparison, the
surface tension depression is also visible but much less pronounced for octyl-OS/AS. Overall, we
proposed that solubility limit and surface tension reduction may both contribute to the observed
discrepancy between $\kappa_{gf}$ and $\kappa_{ccn}$ values for octyl-OS aerosol.
**4. Conclusions**

Organosulfates (OS) may contribute significantly to secondary organic aerosols in various

locations over the globe; however, their hygroscopic properties and CCN activities have not been
well understood. In this work, three complementary techniques, including a vapor sorption
analyzer (VSA), a hygroscopicity tandem differential mobility analyzer (H-TDMA) and a cloud
condensation nuclei counter (CCNc), were employed to investigate interactions of several OS with
water vapor under sub- and supersaturated conditions, trying to get a comprehensive picture of
their hygroscopic properties and CCN activities.

VSA was used to measure mass change of OS samples with RH (0-90%). Obvious

deliquescence was found for sodium methyl sulfate (methyl-OS), sodium ethyl sulfate (ethyl-OS),
sodium octyl sulfate (octyl-OS) and potassium hydroxyacetone sulfate, and their mass growth
factors at 90% RH were determined to be 3.652±0.064, 3.575±0.024, 1.591±0.004 and
2.202±0.031, respectively. No significant water uptake were observed up to 90% RH for other OS
compounds examined, including sodium dodecyl sulfate, potassium 3-hydroxy phenyl sulfate,
potassium benzyl sulfate, potassium 2-methyl benzyl sulfate, potassium 3-methyl benzyl sulfate,



potassium 2,4-dimethyl benzyl sulfate and potassium 3,5-dimethyl benzyl sulfate. Hygroscopic
properties of methyl-, ethyl- and octyl-OS aerosols were also studied using H-TDMA, which
measured mobility diameters of aerosol particles as a function of RH. Continuous hygroscopic
growth was observed for methyl-, ethyl- and octyl-OS aerosols, and their growth factors at 90%
RH were measured to be 1.83±0.03, 1.79±0.02 and 1.21±0.02.
We further investigated CCN activities of methyl-, ethyl- and octyl-OS aerosols, and their
single hygroscopicity parameters, $\kappa_{ccn}$, were determined to be 0.459±0.021, 0.397±0.010 and
0.206±0.008, respectively. For methyl- and ethyl-OS aerosols, single hygroscopicity parameters
derived from CCN activities ($\kappa_{ccn}$) agree reasonably well with those derived from H-TDMA
measurements ($\kappa_{gf}$). However, $\kappa_{ccn}$ was found to be significantly larger than $\kappa_{gf}$ for octyl-OS, and
we show that solubility limit and surface tension reduction may both contribute to such
discrepancy observed.

**Data availability.** Data used in this paper can be found in the main text or supplement.
**Competing interests.** The authors declare that they have no conflict of interest.
**Author contribution.** Mingjin Tang conceived this work; Ru-Jin Huang, Yuqing Zhang, Xiang
Ding and Xinming Wang chose and provided samples investigated in this work; Chao Peng,
Lanxiadi Chen, Yuqing Zhang and Xiang Ding conducted VSA measurements; Chao Peng,
Weigang Wang and Maofa Ge conducted H-TDMA measurements; Patricia N. Razafindrambinina,
Kotiba A. Malek and Akua A. Asa-Awuku conducted CCN activity measurements; Chao Peng,
Patricia N. Razafindrambinina, Kotiba A. Malek, Akua A. Asa-Awuku and Mingjin Tang analyzed
the data and prepared the manuscript with contribution from all the other coauthors.



**Financial support**

This work was funded by National Natural Science Foundation of China (91744204), China Postdoctoral Science Foundation (2020M682931), State Key Laboratory of Loess and Quaternary Geology (SKLLQG1921), Guangdong Foundation for Program of Science and Technology Research (2017B030314057, 2019B121205006 and 2020B1212060053), Guangdong Science and Technology Department (2017GC010501) and CAS Pioneer Hundred Talents program.

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
