# Peer review of "Interactions of organosulfates with water vapor under sub- and supersaturated"

_Atmospheric Chemistry and Physics, 2021_

## Author Comment (AC1)

Comments by referees are in blue.

Our replies are in black.

Changes to the manuscript are highlighted in red both here and in the revised manuscript.

**Reply to referee #1**

This work measures the hygroscopic properties and cloud condensation nucleation (CCN) activities of model organosulfate aerosols and allow us to better understand the environmental fates and impacts of these compounds. This work fits the scope of Atmospheric Chemistry and Physics. I have the following comments and suggestions for the authors' consideration.

**Reply:** We would like to thank referee #1 for reviewing our manuscript and recommending it for publication after revision. His/her comments, which helped us largely improve our manuscript, have been carefully addressed in our revision, as detailed below.

**Comments:**

1. Line 72, "As OS are ubiquitous and abundant in the troposphere, it is important to understand their hygroscopic properties and CCN activities in order to assess their environmental and climatic effects." As mentioned by the authors, the atmospheric abundance of organosulfates investigated in this work is in order of ng/m$^3$. Can the authors further elaborate or comment the atmospheric significances of these investigated organosulfates? For instance, how the presence of these organosulfates would affect the physiochemical properties of atmospheric aerosols such as surface tension, hygroscopicity and CCN studied in this work?

**Reply:** We understand the referee's concern. In fact, for each single organic compound, its contribution to aerosol mass concentrations is small; however, it is still valuable to investigate physicochemical properties of these compounds. Furthermore, the mass fraction of OS in organic aerosols can reach up to 30%, as we pointed out in our original manuscript (line 57-58). To address this comments, in the revised manuscript (line 75-77) we have expanded the sentence to underscore the importance of OS: "As OS are ubiquitous in the troposphere, it is important to understand their hygroscopic properties and CCN activities in order to assess their environmental and climatic effects (Kanakidou et al., 2005; Moise et al., 2015; Tang et al., 2016; Tang et al., 2019a), especially considering that OS could contribute up to 30% of total mass of organic aerosols in the troposphere (Surratt et al., 2008; Tolocka and Turpin, 2012; Liao et al., 2015)."

2. Line 127, "As shown in Figure 2, the mass of OS samples at different RH was determined by the VSA using the following method." In figure 2, only the results of sodium methyl sulfate were shown. What would be the VSA results for other species?

**Reply:** Figure 2 was used as an example (methyl-OS) to show the raw data of our VSA measurements, and thus it is unnecessary and difficult to show raw data for other species. In addition, the VSA results for all species investigated in this work have been summarized in Section 3.1 and Table 1.

3. Line 197, "the increase in sample mass at 50% and 60% RH may be because ethyl-OS were partially deliquesced at this stage." What causes the ethyl-OS partially deliquesce? Why other OSs do not deliquesce partially?

**Reply:** To be honest, we do not have a definite answer to this question. Therefore, similar to many previous studies, we attributed this to partial deliquescence.

4. Line 276, "Figure 4 shows that obvious deliquescence transitions were observed for methyl-, ethyl-, and octyl-OS in the VSA experiments; in contrast, as revealed by Figure 5, continuous hygroscopic growth without obvious phase transitions was observed for methyl-, ethyl- and octyl-OS aerosol particles in H-TDMA measurements, suggesting that these aerosol particles may exist in amorphous state." Can the authors discuss why methyl OS, ethyl-OS and octyl OS exhibited different deliquescence behaviours in H-TDMA and VSA measurements?

**Reply:** The different deliquescence behaviors in H-TDMA and VSA measurements were due to different states of samples used in the two experiments. In the revised manuscript (Line 281-285) we have expanded the two sentences to provide further explanation: "Figure 4 shows that obvious deliquescence transitions were observed for methyl-, ethyl-, and octyl-OS in the VSA experiments, because samples used in VSA experiments may be crystalline salts; in contrast, as revealed by Figure 5, continuous hygroscopic growth without obvious phase transitions was observed for methyl-, ethyl- and octyl-OS aerosol particles in H-TDMA measurements, suggesting that these aerosol particles which were produced by drying aqueous droplets to <5% RH may exist in amorphous state."

5. Figure 6, at the same RH the GFs of methyl-, ethyl- and octyl-OS derived from VSA measurements were found to be consistently smaller than those measured using H-TDMA. Can the authors elaborate this observation?

**Reply:** To address this comment, we have added one sentence in the revised manuscript (Line 303-305) to explain our observation: "The small but systematical differences between VSA and H-TDMA results, as evident from Figure 6, could stem from volume additivity assumption used to convert mass growth to diameter growth, uncertainties in OS densities, and DMA sizing errors."

6. Line 351, "Such underestimation at 70% and 75% RH is likely to due to that inorganic compounds (AS, in our work) may dissolve partially in the organics/water solution (which can be formed at much lower RH due to continuous water uptake of organics) before the mixed particle is completely deliquesced (Svenningsson et al., 2006; Zardini et al., 2008; Wu et al., 2011);" Can the authors comment if the amount of AS partially dissolved could be estimated based on their assumption and the hygroscopic data?

**Reply:** The difference between predicted and measured GF at a given RH (70% or 75%) was due to water associated with partially dissolved AS, and subsequently we can estimate the amount of AS partially dissolved. In the revised manuscript (line 365-367) we have added one sentence to state the amount of partially dissolved AS at 70% RH: "For example, the ratios of partially dissolved AS to total AS at 70% RH were estimated to be 0.95, 0.85 and 0.49 for methyl-OS/AS, ethyl-OS/AS, and octyl-OS/AS mixtures with a mass ratio of 1:1."

7. Line 392, "No significant difference was observed between $\kappa_{gf}$ and $\kappa_{ccn}$ for five types of aerosol particles, and the relative differences between $\kappa_{ccn}$ and $\kappa_{gf}$ values do not exceed 25%. However, octyl-OS appears to be an exception, and the average $\kappa_{ccn}$ value (0.206) was ~1.4 times larger than the average $\kappa_{gf}$ value (0.086)." Given the differences between $\kappa_{gf}$ and $\kappa_{ccn}$ were larger than the error bars (Figure 8), can the authors discuss why they would like to claim there is no significant difference between $\kappa_{gf}$ and $\kappa_{ccn}$.

**Reply:** We agree with the referee, and have revised this paragraph (Line 404-411) in the revised manuscript accordingly: "Figure 8 compares $\kappa_{ccn}$ and $\kappa_{gf}$ values for the six types of aerosol particles examined. For pure OS, $\kappa_{ccn}$ of methyl-OS (0.459±0.021) and ethyl-OS (0.397±0.010) were smaller than their $\kappa_{gf}$ values (0.537-0.604 and 0.505-0.548), but the relative differences do not exceed 25%. Such a difference (<25%) may not be significant if all the uncertainties associated with deriving $\kappa$ from measured hygroscopic growth and CCN activities (Petters and Kreidenweis, 2007). Octyl-OS appears to be an exception, and the average $\kappa_{ccn}$ (0.206) was ~2.4 times larger than the average $\kappa_{gf}$ (0.086). In addition, no significant difference was observed between $\kappa_{ccn}$ and $\kappa_{gf}$ for all the alkyl-OS/AS mixed aerosols."

8. Line 403, "Petters and Kreidenweis. (2008) demonstrated that cloud droplet activation was highly sensitive to the solubility for sparingly soluble compounds in the range of $5 \times 10^{-4}$-$2 \times 10^{-1}$, expressed as volume of solute per unit volume of water (Petters and Kreidenweis, 2008). Compared to the highly soluble methyl- and ethyl-OS, the solubility of octyl-OS ($8.43 \times 10^{-4}$-$4.26 \times 10^{-2}$) (Chemistry Dashboard, 2021) is rather limited." What are the estimated water solubility for methyl- OS and ethyl-OS? Are they highly soluble or sparingly soluble?

**Reply:** The two compounds are highly soluble, and in the revised manuscript (Line 422) we have provided their solubilities: "Compared to the highly soluble methyl- and ethyl-OS (their solubilities are 0.127-0.219 and 0.075-0.151), the solubility of octyl-OS ($8.43 \times 10^{-4}$-$4.26 \times 10^{-2}$) (Chemistry Dashboard, 2021) is rather limited..."

9. Line 408, "incomplete dissolution at subsaturated condition in H-TDMA measurements may lead to underestimation of $\kappa_{gf}$ values for octyl-OS; as a result, the solubility limit may explain the observed difference between $\kappa_{gf}$ and $\kappa_{ccn}$ for octyl-OS." As mentioned by the authors, shall the authors estimate the $\kappa_{gf}$ values for octyl-OS with solubility limit and surface tension correction?

**Reply:** As pointed out correctly by the referee, it will be very nice to assess the contribution of solubility limit and surface tension reduction to the observed difference between $\kappa_{gf}$ and $\kappa_{ccn}$. Nevertheless, this needs sophisticated numerical models and may be beyond the scope of our manuscript. We have added one sentence in the revised manuscript (line 434-436) to acknowledge this caveat: "We note that some numerical models (Petters and Kreidenweis, 2008; Petters and Kreidenweis, 2013; Riipinen et al., 2015) are available to quantitatively assess contribution of solubility limit and surface tension reduction to the discrepancy between $\kappa_{gf}$ and $\kappa_{ccn}$."

---

## Author Comment (AC2)

Comments by referees are in blue.

Our replies are in black.

Changes to the manuscript are highlighted in red both here and in the revised manuscript.

**Reply to referee #2**

The paper by Peng et al. addresses gap knowledge of organosulphates hygroscopic properties and CCN activity. Organosulphate hygroscopic properties have not been systematically studied to date and as such paper is a significant contribution to atmospheric science. The paper is generally well written although a moderate revision is needed to meet the publication standard of ACP.

**Reply:** We would like to thank referee #2 for reviewing our manuscript and recommending it for publication after revision. His/her comments, which helped us largely improve our manuscript, have been carefully addressed in our revision, as detailed below.

1. Line 36. Maintain three significant digits for consistency. Even the third digit of the value is imprecise, because the standard error is changing the second digit.

**Reply:** Thanks for the suggestion, we have made them consistent in our revised manuscript.

2. Line 42. Quantitative numbers needed to illustrate qualitative terms like "reasonably well", especially in the Abstract.

**Reply:** In response to this comment, we have added quantitative numbers in the revised manuscript (Line 42-43): "For methyl- and ethyl-OS aerosols, $\kappa_{ccn}$ values agree reasonably well with those derived from H-TDMA measurements ($\kappa_{gf}$) with relative differences being smaller than 25%, whereas $\kappa_{ccn}$ was found to be ~2.4 times larger than $\kappa_{gf}$ for octyl-OS, likely due to both solubility limit and surface tension reduction."

3. Line 157. delta GF should have reflected Kelvin effect which was not negligible for 100nm particles. The authors could use kappa method in estimating Kelvin effect against e.g., 300nm particle were Kelvin effect would be immeasurable.

**Reply:** We did calculations in our previous work, and the Kelvin effect is negligible for 100 nm particles. In the revised manuscript (line 402-403) we have added one sentence for further clarification: "Eq. (6) does not take into account the Kelvin effect as the effect is small for 100 nm particles (Tang et al., 2016)."

4. Figure 3. Is Figure 3 meant for illustration purposes or is the result? It is unnecessary for the former and if for the latter it should be presented in terms of $SS_{crit}$ as a function of $D_{crit}$ along with ammonium sulphate.

**Reply:** Figure 3 is used for illustration. We fully understand the referee's concern, but we feel that it is necessary for colleagues who are not very familiar with CCN measurements. Therefore, we would like to keep it in the manuscript.

5. Line 195. It is difficult to judge the significance of 11% without uncertainty error bars. Could it be due to physical spatial arrangement of 1mg mass lump?

**Reply:** In fact, errors bars are included in Figure 4, but they are too small to be visible. Our VSA can easily detect a relative mass change of <1%, and a change of 11% is significant. In addition, physical spatial arrangement would lead to change in morphology but not mass.

6. Line 212. Measurable, not obvious. Nothing is obvious in scientific experiment.

**Reply:** That is right, and we have corrected it in the revised manuscript (Line 216).

7. Line 214. suggesting the occurrence of ...

**Reply:** That is right, and we have corrected it in the revised manuscript (Line 218).

8. Line 218. Interestingly, that in this case the authors discount 10-20% increase, contrary to ethyl-OS increase of 11%, mentioned earlier.

**Reply:** The 11% increase for ethyl-OS is reproducible and reliable. The 10-20% increase for the other six potassium organosulfates was only occasionally observed (only for a few experiments) and not reproducible, probably because these chemicals we synthesized contained significant amounts of impurities and were not homogeneous.

9. Table 1. Maintain three significant digits throughout.

**Reply:** Thanks for the suggestion, we have made them consistent in our revised manuscript.

10. Figure 5. I believe that a) and b) were split due to methyl and ethyl OS being similar and partly overlapping, but it is exactly for the same reason they should be on the same graph and if a single graph was bigger it would exhibit those differences clearly.

**Reply:** As suggested by referee #2, the two panels have been merged into one panel in the revised manuscript.

11. Line 254. DMA sizing precision is at best 5% (Wiedensohler et al. 2012, AMT) and, consequently, 7% of the two DMAs. Clearly 8-9% difference can be attributed to sizing uncertainty of different HTDMA systems.

**Reply:** Thanks for the suggestion, we have modified this sentence in the revised manuscript (Line 257-258): "As DMA sizing typically has a relative uncertainty of 5-7% (Wiedensohler et al., 2012), our measured GFs..."

12. Line 297. That is understandable as the bulk material is present in large lump of mass (1mg is huge when compared to single particle). In order for VSA to represent microscopic particles, one should use tiny amount of substance spread as e.g., 100nm film, which is challenging and impractical, thereby limiting the usefulness of VSA for atmospherically relevant studies.

**Reply:** We respect but do not agree with the referee. In fact, many techniques which examine bulk materials provide important data to understand hygroscopicity of aerosol particles, as discussed in a recent review on aerosol hygroscopicity measurement techniques (Tang et al., 2019). Furthermore, in the last few years we have used our VSA instrument to investigate hygroscopic properties of a number of materials with atmospheric relevance, and published several peer-reviewed papers.

13. Line 379. The authors should emphasize that reduced hygroscopicity was measured in supersaturated conditions while in subsaturated conditions hygroscopicity was higher as revealed by HTDMA (e.g. 60-70%).

**Reply:** In fact, a similar trend was also observed for hygroscopicity measured under subsaturated conditions. In the revised manuscript (line 389-390) we have added one sentence to underscore it: "…and this suggests that the addition of hydrophobic hydrocarbon functional groups to OS reduced their hygroscopicity. Decrease in hygroscopicity of OS compounds with the increase in the number of carbon atoms was also observed under subsaturated conditions (Section 3.2)."

14. Line 416. ...but much less pronounced for a mixture octyl-OS/AS. ("mixture" should be emphasized)

**Reply:** That is right, and we have corrected it in the revised manuscript (Line 432): "...but much less pronounced for octyl-OS/AS mixed aerosol".

15. Line 417. What about the discrepancy of methyl and ethyl-OS despite both being very soluble? That should be noted and discussed, especially that their kappa GF are higher than kappa CCN.

**Reply:** We agree with the referee that some discrepancies between $\kappa_{gf}$ and $\kappa_{ccn}$ were observed for methyl- and ethyl-OS. On the other hand, the difference between $\kappa_{gf}$ and $\kappa_{ccn}$ was <25% for these two compounds. As pointed out by Petters and Kreidenweis (2007), if $\kappa$ values vary by <30%,

the difference may not be significant when taking into account the uncertainties associated with deriving $\kappa$ from measured grow factors and CCN activities.

To response to this comment, in the revised manuscript (Line 404-411) we have compared $\kappa_{gf}$ and $\kappa_{ccn}$ values for each compound: "Figure 8 compares $\kappa_{ccn}$ and $\kappa_{gf}$ values for the six types of aerosol particles examined. For pure OS, $\kappa_{ccn}$ of methyl-OS (0.459±0.021) and ethyl-OS (0.397±0.010) were smaller than their $\kappa_{gf}$ values (0.537-0.604 and 0.505-0.548), but the relative differences do not exceed 25%. Such a difference (<25%) may not be significant if all the uncertainties associated with deriving $\kappa$ from measured hygroscopic growth and CCN activities (Petters and Kreidenweis, 2007). Octyl-OS appears to be an exception, and the average $\kappa_{ccn}$ (0.206) was ~2.4 times larger than the average $\kappa_{gf}$ (0.086). In addition, no significant difference was observed between $\kappa_{ccn}$ and $\kappa_{gf}$ for all the alkyl-OS/AS mixed aerosols."